# Relationships between the Clinical Test Results and Neurophysiological Findings in Patients with Thoracic Outlet Syndrome

**DOI:** 10.3390/bioengineering9100598

**Published:** 2022-10-21

**Authors:** Agata Maria Kaczmarek, Juliusz Huber, Katarzyna Leszczyńska, Paulina Wietrzak, Katarzyna Kaczmarek

**Affiliations:** Department of Pathophysiology of Locomotor Organs, University of Medical Sciences, 28 Czerwca 1956 No 135/147, 61-545 Poznań, Poland

**Keywords:** thoracic outlet syndrome, neurophysiological diagnostics, motor evoked potentials, electromyography, ischemic test

## Abstract

A thoracic outlet syndrome (TOS) is the type of brachial plexus disorder most difficult to objectively assess using a clinical examination and differential diagnosis. Its symptoms can be frequently misdiagnosed, especially among others with cervical disc-root conflicts, plexopathies, and peripheral neuropathies. In this study, we aim to identify the correlations between positive Doppler ultrasonography results indicating pathological changes in the subclavian flow velocity, clinical tests, and chosen clinical neurophysiology recordings as proposed alternative or supplementary diagnostic tools for evaluating TOS patients. Sixty TOS patients with positive Doppler ultrasonography and Roos test results and sixty healthy people as a control group were bilaterally examined, and the results were compared. Pain intensity was assessed using a visual analogue scale (VAS). Sensory perceptions within C4–C8 dermatomes were assessed with Von Frey filament (FvF) tests. The activity of motor units in the proximal and distal muscles of the upper extremities was evaluated using surface electromyography (sEMG) during maximal contractions before and after a provocative raised hands test (RHT). An electroneurography (ENG) was used to evaluate the transmission of nerve impulses peripherally. Motor evoked potential (MEP) recordings, induced by the over-vertebral magnetic stimulation of the C5–C7 neuromeres, were used to examine motor transmissions from the cervical motor centres to the upper extremities muscles. The results revealed a relationship between positive Doppler test scores and pathological changes in the subclavian flow velocity through the results of the following diagnostic tools: a VAS score of 1.9 was detected on average, superficial sensory perception abnormalities were found in the innervation areas of the ulnar nerves detected by FvF tests, a decrease in the amplitudes of sEMG recordings was seen in distal rather than proximal muscles (especially following the RHT), a decrease in the motor and sensory peripheral transmissions of nerve impulses in the median, ulnar and cutaneous anterobrachial median nerves was seen, as well as MEP amplitudes recorded from the abductor pollicis brevis muscle. The provocative RHT combined with sEMG and MEP recordings can be considered accurate and objective clinical neurophysiology tools that could supplement the commonly used clinical tests. Such an approach may result in a more precise neurogenic TOS diagnostic algorithm.

## 1. Introduction

Thoracic outlet syndrome (TOS) describes a group of conditions characterized by the compression of the neurovascular bundle, which includes the brachial plexus, the subclavian artery, the subclavian vein, and the axillary vein [1]. The compression may occur in the fissures of the oblique muscles (scalene triangle), the costoclavicular space, and the thoracic space (subcoracoid space). Based on the feedback reported by patients, three types of TOS have been distinguished, i.e., neurogenic, venous, and arterial [2,3]. Neurogenic TOS symptoms include soreness, numbness, and weakness in the upper extremities, pain in the cervical spine, and headaches radiating to the occipital area. These symptoms are reproducibly aggravated by any activity that requires arm elevation [4]. Neurogenic TOS is the most common type of TOS, and according to the description by Hu et al., it covers 90% of TOS cases [5]. Venous TOS is the second most common type of TOS, representing approximately 3% of cases, which manifests itself in lividity and pain, muscle weakness in the upper extremities, and arm swelling, subsequently increasing over a long period of time [6,7]. The paraesthesia symptom is caused by ischemia rather than the result of nerve compression. Arterial TOS is quite rare (approximately 1% of cases) [6,8,9] and is characterized by pain, pallor, coldness of the skin, and paraesthesia. The symptoms of arterial TOS are caused by the compression of the subclavian artery (often through the cervical rib) or by blockages to the blood flow in the venae. Pressure in the subclavian artery in the neck can result in post-stenotic dilation, turbulent flow, and the formation of blood clots. A characteristic feature of all types of TOS is the presence of symptoms, usually when the upper extremities are raised or abducted, either uni- or bilaterally [3,6,9].

If the nerves are compressed in TOS patients, pain can radiate along the ulnar aspect of the forearm into the IV and V fingers, and muscle weakness can descend into the distal part of the upper extremity [10]. Additional symptoms may include paraesthesia, and sensory loss in the area of the innervated ulnar nerve, muscle weakness, and often, one-sided Raynaud’s phenomenon [9]. From the hereinabove mentioned symptoms, it appears that clinical studies that use patient-reported outcomes may mislead a final diagnosis, such as cervical disc-root conflicts, plexopathies, and peripheral neuropathies. Therefore, clinical practitioners are looking for clinical and supplementary functional tests to precisely diagnose TOS.

In the clinical diagnostic process, the most common tests include Adson’s test, Allen’s test, Halstead’s test, and the Roos test, which is one of the most widely used tests for evaluating TOS [11,12]. The sensitivities of these tests are different in TOS detection as, for example, Adson’s test has an estimated 85% sensitivity [11,12,13]. There is a general agreement that supports the effectiveness of Duplex and Doppler ultrasonography, and a low percentage of false positive TOS diagnoses has been documented [5,13,14].

A neurophysiological examination plays a supplementary role in clinical TOS diagnosis, where an electroneurographical examination (ENG), based on the electrical stimulation of the upper extremity nerve branches, is the most commonly used examination [13]. In the case of TOS patients, an ENG reveals abnormalities in the transmission of neural impulses in the motor fibres of the median nerve more than in the ulnar nerve (recorded from the muscles as the M- and F-waves) [15,16] and sensory fibres (through recordings of the SCV potential and sensory conduction velocity studies) of the ulnar nerve more than in the median nerve, either uni- or bilaterally [17,18]. The evaluation of the medial anterobrachial cutaneous sensory response has been found to be a fairly reliable technique for confirming a neurogenic thoracic outlet syndrome diagnosis [19,20]. ENG recordings of evoked potentials are induced by the electrical stimulation of nerves on their anatomical course in patients with suspected TOS, which could show abnormalities in the impulse transmissions along the entire length of the tested branch, with a preference from the Erb’s point level. However, there is no gold standard for the diagnosis of TOS syndrome referred to by the International Federation of Clinical Neurophysiology [21,22]. The effectiveness of magnetic stimulation-induced and motor-evoked potentials (MEPs) to confirm a TOS diagnosis is inconclusive [23] and only speculative. A review of the literature indicates that neurophysiological tests seem to be inferior compared with nerve ultrasound imaging in TOS confirmation, and high-resolution nerve imagining has been found to be valuable for diagnosing patients with neurogenic TOS [24].

There is general agreement that a brachial plexus disorder is the most difficult TOS to be objectively assessed during clinical examinations and differential diagnosis [14]. In addition, it can be misdiagnosed, and a differential diagnosis is challenging due to the possibility of the simultaneous coexistence of peripheral nerve compressions in the upper extremity [25]. Depending on the causes of TOS and its pathological advancements, different medical procedures are recommended for treating patients. Positive therapeutic effects of surgical, pharmacological, and physiotherapeutic treatments have been reported [4,26]. Therefore, confirmation of the effectiveness and usefulness of precise neurophysiological tests for the diagnosis of TOS seems to be of the greatest importance [27].

This study aims to identify correlations between clinical test results and neurophysiological findings in patients with confirmed thoracic outlet syndrome. We compared the neurophysiological test results recorded for patients with a diagnosed TOS to those of healthy people. Moreover, we presented the complete scheme of clinical neurophysiology tests that contribute to an objective diagnosis of TOS, especially during the provocative elevation of arms (raised hands test—RHT).

## 2. Materials and Methods

### 2.1. Study Design, Participants, and Clinical Evaluation

The same clinical neurophysiology tests were conducted once on 60 patients (in the study group), and 60 healthy volunteer subjects were identified (as the control group) with negative Roos test results, and no Doppler test was required to obtain reference values. The characteristics of the subjects enrolled in the study are presented in Table 1. The studies were performed between 2017 and 2022. Ethical considerations were in agreement with the Helsinki Declaration. Approval was also received from the Bioethical Committee of the University of Medical Sciences in Poznań, Poland (including for the studies on healthy people, no. 554/2017). All patients understood that there was no financial benefit from participation, and they signed a written consent form for voluntary participation in the study.

The exclusion criteria included head injury, stroke, epilepsy episodes, mental disorders, cardiovascular disease, having a pacemaker or cochlear implant, pregnancy, oncological episodes, inflammatory disease, myelopathy, degenerative changes in the cervical spine (after verification on CT or MRI scans), peripheral neuropathies from the level of the arm (verified in electroneurographical studies), and COVID-19-related symptoms.

Patients with clinically confirmed TOS (including neuroimaging studies) were qualified for the clinical neurophysiology diagnostics unit by the same experienced specialist neurologist. The inclusion criteria were as pain in the upper extremities or neck area and suspected TOS, i.e., a positive Roos test result and positive Doppler imaging (confirming the TOS pathology) were always required. According to studies by Longley et al. and Wadhwani et al., the flow velocity in the subclavian artery is 50–100cm/s in normal conditions. Therefore, any increase in the velocity index >2 times was considered to be a symptom of significant compression and corresponded to a stenosis >50%, which was assessed as a positive Doppler test result [28,29].

Moreover, the sensory perception studies, bilaterally towards an analogue diagnosis, were assessed according to the dermatomal scheme for the covering innervation of the musculocutaneous, median, and ulnar nerves based on the tactile method with Von Frey’s filaments (Semmes–Weinstein monofilaments) [30,31]. Uni- or bilateral pain was evaluated based on a patient-reported 10-point visual analogue scale (VAS) score [32].

The study design includes the presentation only of results from the symptomatic (or more symptomatic) upper extremities in TOS patients in comparison to the results from the right extremity of the control subjects (59 healthy subjects were right-handed, 1 subject was left-handed). In healthy volunteers (the controls), we preliminary compared the neurophysiological study parameters of the recordings and did not find significant differences between the right and the left side, although the general prevalence of the right hand has been noticed in the analysis of amplitudes from the sEMG recordings. The subjects from the group of healthy volunteers were recruited randomly from a population of more than one-hundred studied people and were evaluated with clinical and neurophysiological tests every five years for the purpose of obtaining the normative parameters.

### 2.2. Neurophysiological Testing

The neurophysiological testing was performed using a Keypoint System (Medtronic A/S, Skovlunde, Denmark), according to the guidelines of The International Federation of Clinical Neurophysiology, in the European Chapter. Figure 1 presents the principles of sequentially performed neurophysiological tests, forming the diagnostic algorithm proposed in this study, which was aimed at facilitating the differential diagnosis of TOS. It consisted of the electromyographic recordings from the distal (Figure 1A) and proximal (Figure 1C) muscles bilaterally (also during the rising and abduction of the arms to evoke temporary ischemia in the subclavian artery, Figure 1B), the electroneurographical stimulation studies of the sensory (Figure 1E,G) and motor (Figure 1D,F) fibres in upper extremity nerve branches, as well as in the evaluation of the total efferent neural transmission studies from the cervical spinal centres to the effectors (Figure 1H).

The surface electromyography (sEMG) recordings aimed to ascertain the contractile properties of the muscle motor units during a maximal contraction attempt lasting 5 s, and were performed bilaterally by the abductor pollicis brevis (APB) muscles before (Figure 1A) and after the ischemic “raised hands test” (RHT) (Figure 1B). A patient raised their hands over their head in half-elbow for 2 min. A decrease in the sEMG amplitude parameter of more than 50% after the RHT (measured in µV minimum–maximum, the peak-to-peak of the recruiting motor unit action potential deflection made reference to the isoelectric line) was ascertained as a positive ischemic test and an indicator of TOS. The RHT during the sEMG recordings was performed twice on each of the subjects. Moreover, the sEMG studies during the maximal contraction attempt comprised bilateral recordings from the abductor digiti minimi (ADM) and the biceps brachii (BIC) muscles (Figure 1D). Standard disposable Ag/AgCl surface electrodes with an active surface of 5 mm^2^ were used, which were placed over the muscle belly, with a reference electrode placed on the distal muscle’s tendon. The ground electrode was located in the nearest vicinity to the recording pair. The upper 10 kHz and lower 20 Hz filters of the recorder were used. During the first stage of the examination, the patient was asked to fully relax the examined muscles and then perform a maximal contraction for 5 s, during which the simultaneous recording took place. Participants were instructed to contract the tested muscle as hard and quickly as possible until the neurophysiologist requested them to finish the attempt. The test was conducted three times, with a one-minute interval resting period between each muscle contraction. The recording with the highest amplitude (in µV) and frequency (in Hz) parameters were chosen for the final analysis. The sEMG recordings were performed at the base time of 80 ms/D and amplification of 20–1000 µV. The outcome measures were the amplitude parameters both for the healthy volunteer subjects and the patients (Figure 1); the methodological and analysis principles of sEMG have been described elsewhere [33,34,35].

The electroneurography (ENG) of the median and ulnar nerves was used for the bilateral detection of changes in the transmission of neural impulses in the motor and sensory peripheral fibres. Following the application of electrical, rectangular pulses with a 0.2 ms duration of 1 Hz and an intensity from 0 to 80 mA delivered from the bipolar stimulating electrodes over the skin along their anatomical passages, the M-waves and F-waves were recorded from the APB and the ADM muscles (Figure 1 F). Recordings of these potentially verified transmissions of neuronal impulses in the peripheral motor fibres and within the C6–C8 ventral spinal roots were taken. During the recordings of sensory conduction velocity studies (SCV potentials) in ENG at the wrist, transmissions in the median and ulnar nerve sensory fibres were analysed after a bipolar stimulation of the second and fifth fingers (Figure 1G). The pairs of surface electrodes recorded the evoked potentials, and the same types of surface electrodes were also used for the sEMG recordings. The recordings were performed at an amplification of 5–5000 µV and a time base of 2–10 ms, and then the normative values recorded in the healthy volunteer subjects were compared with the patients of both groups (Figure 1). The outcome measures were the parameters of amplitudes (in µV) and latencies (in ms) in the M-wave and SCV potential recordings, and F-wave frequencies (no less than 14, during the evoking of 20 positive, successive recordings of M-waves were conducted). 

Electroneurography recordings of the musculocutaneous nerves on both sides following the stimulations of the motor fibres at the Erb’s points bilaterally and the recordings of M-waves from the bicep’s brachial muscles (Figure 1D) in order to verify the peripheral motor innervation from the C5 neuromers, were not expected to be disturbed in the TOS patients. By contrast, the transmission of sensory neural impulses peripherally within the fibres of the cutaneous anterobrachial median nerves, originating from the lower part of the brachial plexus, was expected to express more axonal than demyelinating changes in the SCV recordings following their stimulation at the cubital fossa and recordings along the median aspect of the forearm (Figure 1E). More detailed descriptions of hereinabove mentioned ENG studies are provided in the studies by Kothari et al. [19], Huber et al., and Kaczmarek et al. [35,36].

Following the descriptions from Bryndal et al. [33] and Wincek et al. [34], we used oververtebrally induced motor-evoked potentials (MEPs) by applying a magnetic field to verify the efferent transmissions from the spinal motor centres in the C5–C8 neuromeres to the effectors (recordings from BIC, APB, and ADM muscles) (Figure 1H). A MagPro R30 (Medtronic A/S, Skovlunde, Denmark) was used to generate the motor-evoked potentials a single stimulus was delivered with a circular coil (C-100, 12 cm in diameter). The recordings were performed at an amplification of 100–5000 µV and a time base of 5–20 ms and were compared with the normative values recorded in the healthy volunteer subjects (Figure 1). The outcome measures were the parameters of amplitudes (in µV) and latencies (in ms) of the MEPs.

### 2.3. Statistical Analysis

Statistical data were calculated with Statistica 13.3 software (StatSoft, Kraków, Poland). Descriptive statistics included minimal and maximal values (range), and mean and standard deviations (SD) for measurable values, while the frequency of incidence was ascertained for the categorical variables. The overlapping of the positive test results from the clinical studies, imaging study results, and results of the sEMG studies with positive RHTs were expressed as the percentages for 60 studied TOS patients. The cumulative data from the symptomatic side (or more symptomatic side in cases where TOS symptoms were detected on both sides) were used, taking also into consideration the possibility of the symptom reversal phenomenon. The results from all neurophysiological tests performed in patients with TOS were also calculated from the group of healthy subjects (control group) to achieve the normative parameters used for the comparison of the health status between the patients and the controls. The Shapiro–Wilk test was conducted to assess the normality of distributions; however, a large number of patients and healthy volunteers (over 50 individuals in each group) enabled the use of simple Student’s t-test instead of nonparametric tests in cases of the normality absence. The differences between the neurophysiological parameters before and after RHT were evaluated and compared as dependent groups with a dependent Student’s *t*-test (a paired difference *t*-test), whereas the differences between the patients and healthy subjects were calculated using an independent Student’s *t*-test. *P*-values of less than 0.05 were considered to be statistically significant. Spearman’s rank correlation (r_s_) was used to assess the relationship between clinical and neurophysiological studies of TOS patients. *P*-values <0.05 were assumed for rank correlation to be statistically significant.

At the beginning of the pilot studies, the estimation of the sample size was based on the analysis of results obtained in the first 20 subjects and the difference between the amplitudes of sEMG recordings before and after RHT tests. The estimated sample size was 40, but we decided to increase this number to avoid possible dropouts. The rationale for performing the comparative studies on an equal number of 60 patients and 60 healthy subjects was the high, compatible number of participants for the purpose of reliable statistical analysis.

## 3. Results

The patients and the healthy volunteer subjects did not differ significantly in sex, age, height, weight, or BMI (Table 1).

Based on the results of the clinical and Doppler tests, bilateral symptoms of TOS have been identified in 19 subjects while also unilaterally in 41 subjects. For the purposes of the statistical analysis, however, there were finally ascertained to be 29 subjects with predominantly right and 31 subjects with predominantly left TOS symptomatic (or more symptomatic) upper extremities. This data allowed for the gathering and setting of the corresponding results of clinical, Doppler, and neurophysiological studies recorded from the symptomatic side and used for further statistical analysis.

The TOS patients who complained of bilateral pain were assessed using VAS scores, with a mean score of 1.9 (a range from one to six).

We found high percentages of coincidence (95%) between pain symptoms and decreased sensory perceptions of the median and ulnar nerve innervation areas during clinical examinations of the TOS patients (Table 2). In general, we did not find such a high relationship in FvF studies when applied to the sensory innervation of musculocutaneous nerve fibres (only 4 out of 60 patients). Moreover, we observed a coincidence between positive Doppler examination results, and the presence of pain symptoms in VAS (93%), and the lowered amplitude values of sEMG recordings from the APB muscles after RHT (90%).

The sEMG test was performed before and after the RHT, which resulted in a decrease in the recorded amplitude parameter of the TOS patients by about five minutes, on average, to the full recovery of its initial value. The Doppler subclavian artery flow velocity on the symptomatic side ranged from 1.5 m/s to 2.3 m/s (mean 1.7 ± 0.1 SD) in TOS patients (Figure 2).

The data in Table 3 provide evidence of significantly different health statuses between the healthy volunteer subjects and the TOS patients with reference to the performed EMG, ENG, and MEP studies. All the results of the neurophysiological tests performed by patients with the recording sites in the muscles innervated mostly by the median nerve and the inferior trunk of the brachial plexus showed a significant decrease in the amplitude parameters, which proved the neurogenic abnormality of the axonal type.

In TOS patients, the EMG results during the maximal contraction attempt were characterized by a decrease in the amplitude parameter at *p*-values of 0.0001–0.0005 when recorded from the abductor pollicis brevis and abductor digiti minimi muscles but not when recorded from the biceps brachii muscles. Moreover, the same recordings performed in patients after RHT were significantly lower (at *p-*value = 0.0001) than before the provocation test providing evidence of a decrease in muscle motor unit activity following the ischemia effect.

The ENG studies following the stimulation of the motor (M-wave tests) and sensory (SCV tests) fibres of the median and ulnar nerves revealed a significant decrease in the amplitude of the motor-evoked potentials, suggesting an axonal-type injury rather than the demyelination processes or the conduction block phenomenon. The data on the M-wave parameters in the ENG recordings, following the stimulation of the musculocutaneous nerves, did not present any statistical differences between recordings in the TOS patients and the healthy volunteer subjects. On the other hand, a decrease in the amplitude parameters with a simultaneous increase in the latency values in the ENG studies of cutaneous antebrachial median nerve transmissions, provided evidence of statistically significant differences between the healthy volunteer subjects and the TOS patients (*p*-values = 0.0001–0.0002).

It seems that the MEP studies in TOS patients, with a significantly decreased amplitude but no latency parameters (at *p*-values 0.0005 and 0.0006) and with recordings from the APB and ADM muscles but not from the bicep’s brachii muscles, reflect the same axonal changes (detected in ENG) in the efferent transmissions from the level of the cervical spinal motor centre. F-wave responses in the ENG studies were only slightly decreased in incidence following the stimulation of the median nerves, indicating a lack of significant pathological changes in the ventral root impulse transmissions.

The correlation study results of the examined parameters in the clinical and neurophysiological studies of TOS patients are presented in Table 4. We found significant positive correlations between the positive Doppler results and pain symptom detection (in VAS scale) as well as negative abnormalities in the sensory perceptions of the ulnar innervation area (in FvF studies). Moreover, we found negative correlations between the positive Doppler test results, indicating an increase in subclavian flow velocity (m/s) and a decrease in the amplitude parameters of sEMG during maximal contraction recordings, as well as in sEMG during RHT (see also Figure 2). Negative correlations between positive Doppler test results and abnormalities in the motor transmission of neural impulses were recorded in the ENG and MEP studies and were found in the vast majority of the patients (Table 4).

## 4. Discussion

Usually, neurologists, more often than neurosurgeons, orthopaedists, or vascular surgeons, and in the end, rehabilitation specialists are the main physicians who “collide” with the problem of “fainting hands” reported by patients. Depending on the development of pathological symptoms, “objective” complaints from a patient may wrongly direct a physician to a lengthy differential diagnosis based on clinical, manual tests supplemented by using USG, MRI, or CT neuroimaging of the cervical spine and brachial plexus on the symptomatic side, Doppler studies, and in the end, supplementary clinical neurophysiology diagnostics in order to confirm the neurogenic TOS origin. However, overlapping pathological symptoms originating from the consequences of cervical disc-root conflicts, plexopathies, and peripheral neuropathies, may outshine or be misdiagnosed as TOS. If we consider the contemporary attempts of neurophysiologists to verify the usefulness of nerve conduction studies (usually applied in a brachial plexus injury differential diagnosis) in TOS evaluation, the vast majority of tests have to be performed, as well as needle electromyography recordings, from several muscles of the upper extremity [21,22]. Therefore, the primary goal of our study was to correlate the results of clinical and neurophysiological testing by using especially new and non-invasive methods, such as magnetically-induced and over vertebral motor-evoked potentials at the cervical level or surface electromyography, which have increasingly wider applications in clinical neurophysiology studies [31,34,35]. The flow chart of the diagnostic algorithm of the patients proposed in this study was aimed at facilitating the differential diagnosis of TOS, as presented in Figure 3.

This study proves that there is a high correlation between positive Doppler test results confirming TOS and electromyographical findings in the distal upper extremity muscles’ pathology following ischemia evoked by performing the “raised hands test” (RHT). Therefore, we propose sEMG, especially after RHT, as a supplementary, high-sensitivity tool for confirming TOS-related pathologies. Such a novel TOS diagnostic algorithm fulfills the assumption provided by Sanders et al. and Povlsen et al. [3,9], who underlined the importance of raised or abducted positions in the upper extremities to provoke TOS symptoms. Our study also points out the importance of the motor-evoked potential (MEP) diagnostic test that was used to verify abnormalities in the efferent transmissions of neural impulses in the brachial plexus fibres in the case of TOS. This conclusion is based on a high percentage of the coincidence of abnormal MEP results in patients with positive Doppler test results and, up to now, MEPs have not been proposed to be widely used in TOS diagnostics, except in the casuistic study by Haghighi et al. [23]. Similar to the results of our research, they did not reveal any changes to MEP latencies with diminishing amplitudes after applying the dynamic position of the arms.

We intentionally chose sEMG recordings for the distal and proximal muscles in TOS patients. No changes in the bicep’s brachii muscle motor units’ or contractile properties were expected, contrary to recordings from hand muscles innervated by middle and low branches of the brachial plexus, which usually are compressed in patients with neurogenic TOS. Our ENG and sEMG studies confirmed previous observations on neural transmission pathologies in sensory and motor fibres in the medial, ulnar, and cutaneous anterobrachial medial nerves but not in the proximal muscles innervated from C4/C5 neuromeres, including the innervation of musculocutaneous motor fibres [15,16,17,18,23,24]. A significant decrease in the frequency of F-wave recordings and subsequently increased M-F interlatencies following ulnar and median nerve stimulation proved the level of the neural transmission pathology within the area of the spinal C6–C8 ventral root. However, it should be remembered that such secondary to primary TOS abnormalities have been reported in previous neurophysiological studies [15]. M-wave recordings with diminished amplitudes following the stimulation of the ulnar and median nerves on the symptomatic side may indicate that the axonal changes propagate peripherally. The consequences of such pathologies can be clearly seen in the MEP recordings from the hand muscles but not the arm muscles (see Table 3). Deficits in the sensory fibre transmissions of the median and the ulnar peripheral nerves (SCV studies) strictly correlate with the positive tactile FvF test results with reference to the hereinabove mentioned nerve branches.

The most surprising result in the clinical studies was the low VAS pain scores reported by TOS patients (a mean score of 1.9 and a range from 1 to 6). Higher pain intensity was assessed through VAS scores, i.e., 3.6 on average, has been reported by Hwang et al. in neurogenic TOS patients [37]. However, it should be remembered that the VAS is a subjective evaluation tool that depends on a patient’s report, which is considered to be a study limitation.

The proposed algorithm for early clinical neurophysiology diagnostics of neurogenic TOS allows for the early detection of pathological symptoms, including autoimmunological and polyneuropathies of different origins, and by using correlations with clinical test results, the aim is to prevent disease development associated with conservative treatment methods. This suggests that neurophysiological recordings may facilitate the precise differential diagnostics of neurogenic, venous, and arterial TOS. Moreover, consecutive clinical neurophysiological sessions that include non-invasive methods for testing may reduce the need for further conservative or surgical intervention.

One of the study limitations is that the standard needle EMG studies of individual muscles in TOS patients were not performed. However, from the very beginning of the project, our intention was to search for non-invasive clinical neurophysiology methods for evaluating the muscle function in TOS patients, which, among others, include sEMG recordings. Moreover, it would be methodologically difficult and painful to perform the proposed RHT with a needle EMG inserted into the abductor pollicis brevis muscle bilaterally, for example, in the hand position of studied subjects (including the healthy volunteer subjects) as presented in Figure 1B.

## 5. Conclusions

There is a high correlation between the clinical test results and neurophysiological findings in patients with thoracic outlet syndrome. The proposed TOS diagnostic algorithm seems to be a fast and specific set of tools for TOS confirmation and its pathological consequences within the brachial plexus motor fibres. In the diagnosis of TOS, sEMG following RHT and MEP recordings from the distal muscles of the upper extremities are of the greatest relevance.

## Figures and Tables

**Figure 1 bioengineering-09-00598-f001:**
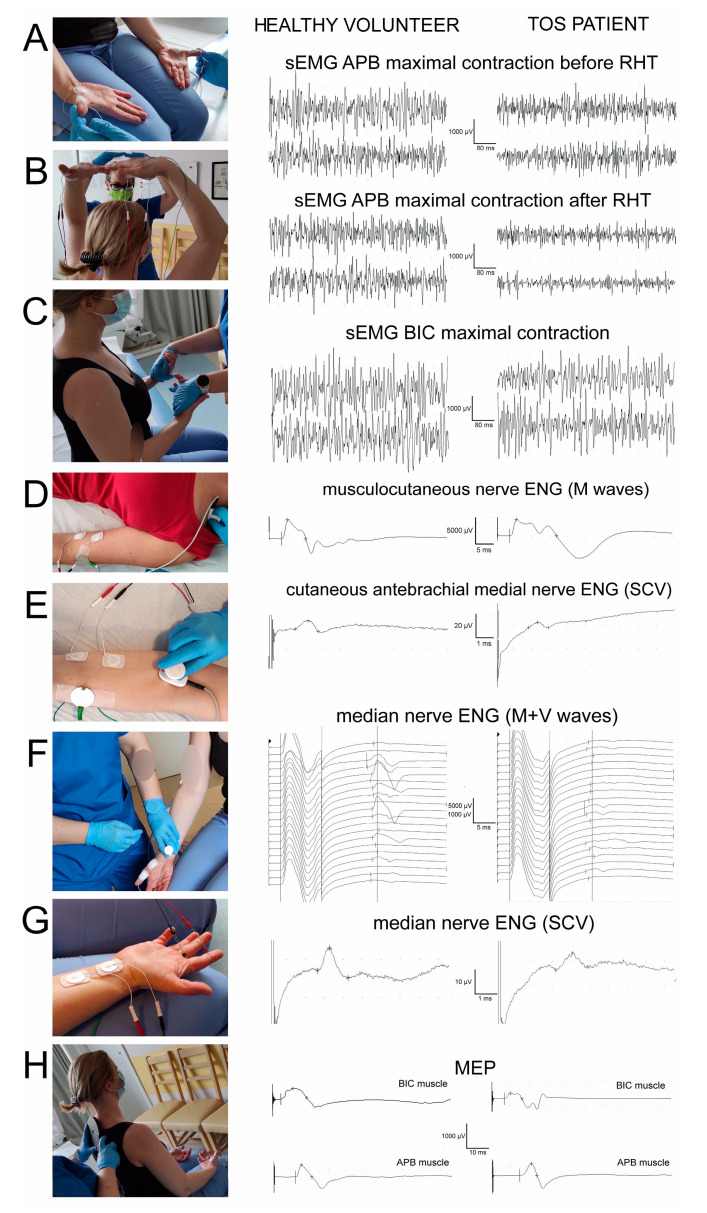
Photographs illustrating the principles of: (**A**–**C**) surface electromyography (sEMG), (**D**–**G**) electroneurography (ENG), and (**H**) motor-evoked potential (MEP) of recordings. (**A**–**H**) Examples of the original, representative recordings of each of the applied neurophysiological tests performed in one of the healthy volunteer subjects and the TOS patients for comparison. Details of sequentially performed neurophysiological tests, forming the diagnostic algorithm proposed in this study, facilitated the differential diagnosis of TOS detection, and are presented in the Materials and Methods section. Calibration bars for different amplifications and time bases are presented.

**Figure 2 bioengineering-09-00598-f002:**
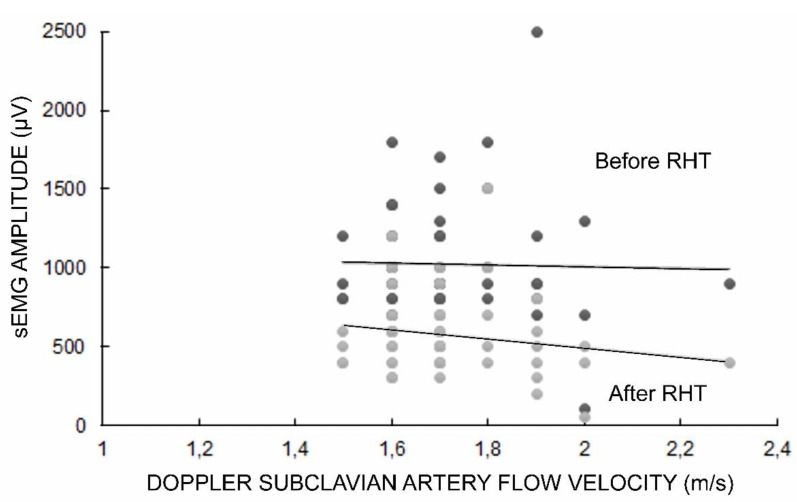
Graphical presentation of relationships between positive Doppler test results (indicating pathological changes in the subclavian flow velocity), and maximal contraction sEMG amplitude recordings before and after RHT tests in TOS patients on the symptomatic side. Some points representing parameters of sEMG vs. Doppler results are overlapped multiple times with their Y-X locations. Black dots refer to results before RHT, while grey dots refer to after RHT.

**Figure 3 bioengineering-09-00598-f003:**
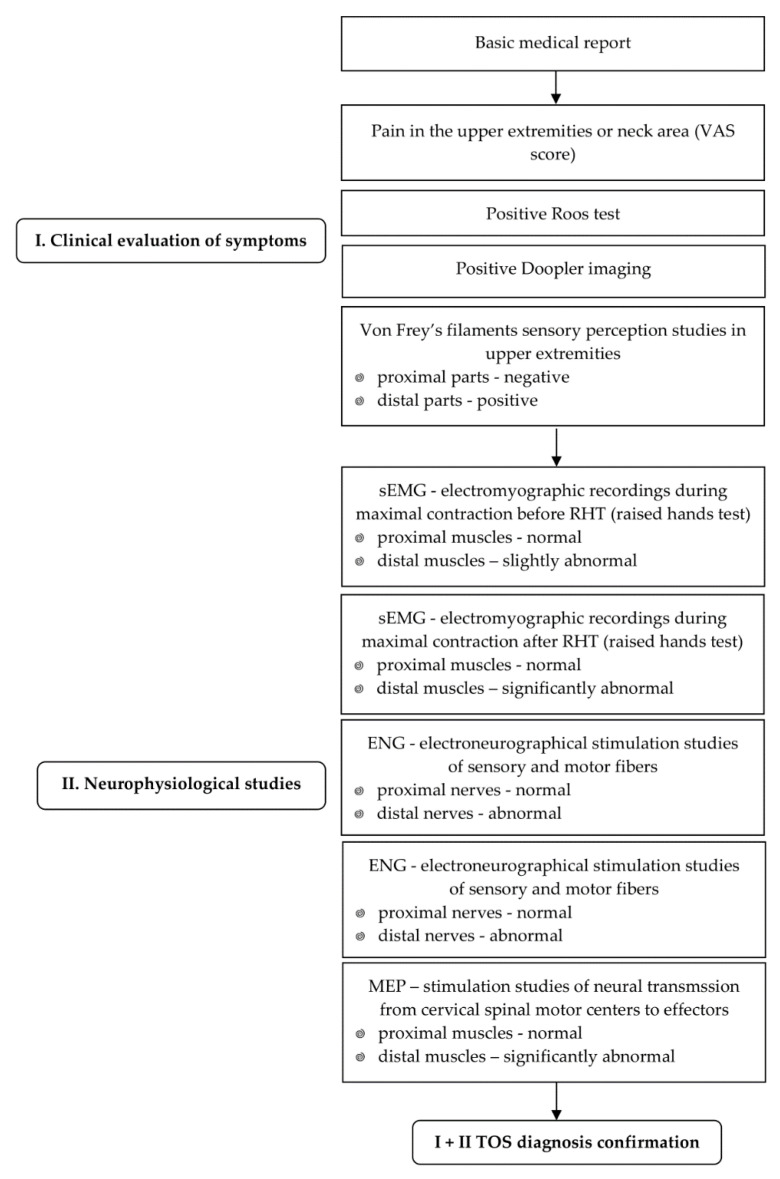
Flow chart of the diagnostic algorithm of the patients with TOS proposed in this study.

**Table 1 bioengineering-09-00598-t001:** The characteristics of the subjects enrolled in the study.

Study Group Variable	Healthy Volunteer Subjects (Control) N = 60 ♀ = 43, ♂ = 17		Patients N = 60 ♀ = 45, ♂ = 15		Control vs. Patients
	Mean ± SD	Min–Max	Mean ± SD	Min–Max	*p*-Value
Age (years)	29.9 ± 7.9	18–50	30.8 ± 7.4	18–52	0.06
Height (cm)	171 ± 5.8	155–180	170 ± 6.3	155–190	0.57
Weight (kg)	57.7 ± 9.9	40–80	57.0 ± 8.9	42–80	0.83
BMI	19.6 ± 2.7	15.2–25.8	19.5 ± 2.8	15.5–28.9	0.67

**Table 2 bioengineering-09-00598-t002:** The overlapping of the positive tests results recorded on the symptomatic side in clinical and neurophysiological examinations of the 60 studied TOS patients.

Positive Tests	Number of Coincidence	Percentage of Coincidence
VAS and FvF median nerve	57	95%
VAS and FvF ulnar nerve	57	95%
VAS and Doppler	56	93%
sEMG APB after RHT and Doppler	54	90%

VAS—pain intensity on the 10-point visual analogue scale, FvF—the study of superficial sensory perception with Von Frey filaments (0 decreased perception and 1 normal), sEMG APB—surface electromyography recorded from abductor pollicis brevis muscles, RHT—raised hands test, Doppler—Doppler ultrasound.

**Table 3 bioengineering-09-00598-t003:** Comparison of the results of neurophysiological tests in healthy subjects (N = 60) and TOS patients (N = 60). The mean values with standard deviations are presented. The most significant *p*-values < 0.05 are marked in bold.

Parameter Recording Site		Healthy Volunteers (Control)	TOS Patients	Control vs. TOS Patients *(p*)
		Right Side	Symptomatic Side	
**sEMG**				
APB bEMG (µV) APB aEMG (µV)		1611.7 ± 290.1 1652.3 ± 275.0	1020.0 ± 360.2 579.1 ± 255.9	**0.0002** **0.0005**
		**Before vs. after (*p*) 0.0001**		
ADM EMG (µV)		1668.3 ± 272.8	1071.6 ± 258.4	**0.0001**
BIC EMG (µV)		2100 ± 611.5	1626.0 ± 320.2	0.009
**ENG—musculocutaneous nerve**				
M-wave	Amplitude (µV)	6033.3 ± 382.7	5941.6 ± 241.8	0.114
	Latency (ms)	5.5 ± 0.4	5.6 ± 0.5	0.128
**ENG—cutaneous anterobrachial median nerve**				
SCV	Amplitude (µV)	23.0 ± 1.9	11.5 ± 2.1	**0.0002**
	Latency (ms)	2.3 ± 0.1	3.5 ± 0.2	**0.0001**
**ENG—median nerve**				
M- wave wrist	Amplitude (µV)	7656.7 ± 1629.5	2186.6 ± 1386.6	**0.0001**
	Latency (ms)	3.5 ± 0.2	3.7 ± 0.3	0.009
SCV- wrist	Amplitude (µV)	16.3 ± 3.0	8.7 ± 4.0	**0.004**
	Latency (ms)	3.4 ± 0.2	3.4 ± 0.2	0.08
F wave [x/20]		16.4 ± 1.5	12.6 ± 3.4	**0.008**
M-F latency [ms]		24.4 ± 1.2	29.2 ± 6.0	**0.009**
**ENG—ulnar nerve**				
M- wave wrist	Amplitude (µV)	7125.0 ± 1543.1	3728.3 ± 1897.0	**0.005**
	Latency (ms)	3.5 ± 0.3	3.4 ± 0.3	0.06
SCV- wrist	Amplitude (µV)	17.0 ± 4.3	7.9 ± 2.9	**0.005**
	Latency (ms)	3.4 ± 0.3	3.4 ± 0.3	0.13
F wave [x/20]		16.4 ± 1.5	12.4 ± 3.2	**0.007**
M-F latency [ms]		24.4 ± 1.2	29.1 ± 6.6	**0.006**
**MEP**				
BIC	Amplitude (µV)	1884.2 ± 244.7	1691.6 ± 258.6	0.01
	Latency (ms)	7.2 ± 0.2	6.7 ± 0.4	0.04
APB	Amplitude (µV)	1524.3 ± 422.1	1065.8 ± 409.0	**0.0005**
	Latency (ms)	16.3 ± 1.2	16.8 ± 1.3	0.08
ADM	Amplitude (µV)	1536.3 ± 432.1	1024.1 ± 388.7	**0.0006**
	Latency (ms)	16.4 ± 1.5	15.4 ± 2.0	0.14

sEMG— surface electromyography, APB—abductor pollicis brevis muscle, bEMG—amplitude before raised hands test, aEMG—amplitude after raised hands test, ADM—abductor digiti minimi muscles, BIC—biceps brachii muscle, ENG—electroneurography, SCV—sensory conduction velocity studies, F-wave (x/20)—frequency of recorded F waves following 20 applied evoking M-wave responses, M-F—value of interlatency between recorded M and F waves, MEP—oververtebrally induced motor-evoked potentials.

**Table 4 bioengineering-09-00598-t004:** Spearman’s rank correlation (r_s_) of the test results obtained in clinical and neurophysiological studies on the symptomatic side in TOS patients. *P-*values < 0.05 are assumed, for rank correlation, to be statistically significant.

Parameter or Test	Symptomatic Side	
	Positive Doppler	
VAS	r_s_	P
	0.75	0.03
Positive Doppler		
FvF ulnar nerve	r_s_	P
	−0.15	0.04
Positive Doppler		
sEMG APB	r_s_	P
	−0.34	0.04
Positive Doppler		
sEMG RHT APB amplitude (µV)	r_s_	P
	−0.28	0.04
Positive Doppler		
ENG M−wave ulnar nerve amplitude (µV)	r_s_	P
	−0.3	0.03
Positive Doppler		
MEP APB amplitudes (µV)	r_s_	P
	−0.1	0.04

Positive Doppler—positive Doppler test results, VAS—positive pain score, FvF—the study with Von Frey filaments of abnormal superficial sensory perception, sEMG APB—amplitude parameters recorded in electromyography, sEMG RHT—surface electromyography amplitude recording after raised hands test, ENG ulnar nerve—M-wave amplitude in electroneurography studies following ulnar nerve stimulation, MEP APB—amplitude parameters in motor evoked potentials.

## Data Availability

All the data generated or analysed during this study were included in this published article.

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
