# Peer review of "Relationships between the Clinical Test Results and Neurophysiological Findings in Patients with Thoracic Outlet Syndrome"

_bioengineering, 2022, doi:10.3390/bioengineering9100598_

Round 1
Reviewer 1 Report
Dear Editor and Authors,
Thank you for asking me on behalf of the journal to review this manuscript titled “Relationships Between the Clinical Test Results and Neurophysiological Findings in Patients with Thoracic Outlet Syndrome” by Dr. Kaczmarek and colleagues from the Department of Pathophysiology of Locomotor Organs at the University of Medical Sciences in Poznań, Poland.
In this study the authors aimed as they report to “identify correlations between clinical test results and neurophysiological findings in patients with confirmed thoracic outlet syndrome”. To do so they compared the neurophysiological test results recorded for patients with a diagnosed TOS and of healthy people.
I do have some comments:
1 1. The abstract seems a bit too long and explanatory – it reads more like the materials and methods section with the listing of all the tests performed. I would suggest a re-write and a reduction in size.
2 2. The introduction is well written, a bit extensive but then it gives a certain level of knowledge and background to the pathology that some readers may need. I am fine with this.
3. The authors do not report in their materials and methods section if this was a prospective study? I am aware it was not/could not be randomized.
4. How were the healthy subjects studied selected?
5. It is good that a power/sample size calculation was performed prior to commencement to see if the number of patients would sufficient to provide statistically meaningful results.
6. That informed consent was obtained by all subjects should be moved and reported along with the ethical approval section (after it!).
7. Table 1 and its associated paragraph should be moved in the results section and not in the materials & methods one.
8. Why are the 17 years of experience of the neurologist important? If he had 20 would it be better? Just say, “an experienced specialist neurologist”!
9. Lines 147 to 157 is more reporting of findings than materials & methods. Maybe it should be simplified for the M&M section and further reported in the results.
10. The results are interesting and well reported.
11. The discussion is also good.
1 12. Good language, clear and understandable. Well illustrated manuscript with clear tables.
In conclusion this is an interesting study with good outcomes and results. It is well reported and really only needs some minor corrections as the methodology and conduct is solid. I wish well to all and take care.
Author Response
Dear Reviewer 1,
Please see the attachment.

Reviewer 2 Report
This is a thorough comparison of various diagnostic tools of thoracic outlet syndrome.
There are a few minor points and suggestions:
- In describing forearm, it may be clearer to use radial or ulnar aspect instead of lateral or medial (which depends on the position of the arm).
“If the nerves are compressed, in TOS patients, pain radiates along the lateral part of 64 the forearm to the IV and V fingers, and muscle weakness descends to the distal part of 65 the upper extremity [10]. “
- Demographic data and hence figure 1 conventionally are best presented in the Result section rather than Methodology section
- Some investigations, such as ultrasound, may be more readily available in regional hospitals than EMG, ENG, MEP. Hence a practical algorithm of tests sequence, as in a flow chart, may be helpful to clinicians in reaching the diagnosis of TOS.
Author Response
Dear Reviewer 2,
Please see the attachment.
